# Evaluation of Optical Genome Mapping in Clinical Genetic Testing of Facioscapulohumeral Muscular Dystrophy

**DOI:** 10.3390/genes14122166

**Published:** 2023-11-30

**Authors:** Anja Kovanda, Luca Lovrečić, Gorazd Rudolf, Ivana Babic Bozovic, Helena Jaklič, Lea Leonardis, Borut Peterlin

**Affiliations:** 1Clinical Institute of Genomic Medicine, University Medical Center, 1000 Ljubljana, Slovenia; anja.kovanda@kclj.si (A.K.);; 2Faculty of Medicine, University of Ljubljana, 1000 Ljubljana, Slovenia; 3Institute of Clinical Neurophysiology, University Medical Center Ljubljana, 1000 Ljubljana, Slovenia

**Keywords:** optical genome mapping, OGM, facioscapulohumeral muscular dystrophy, FSHD1, FSHD2

## Abstract

Facioscapulohumeral muscular dystrophy (FSHD) is the third most common hereditary muscular dystrophy, caused by the contraction of the D4Z4 repeats on the permissive 4qA haplotype on chromosome 4, resulting in the faulty expression of the *DUX4* gene. Traditional diagnostics are based on Southern blotting, a time- and effort-intensive method that can be affected by single nucleotide variants (SNV) and copy number variants (CNV), as well as by the similarity of the D4Z4 repeats located on chromosome 10. We aimed to evaluate optical genome mapping (OGM) as an alternative molecular diagnostic method for the detection of FSHD. We first performed optical genome mapping with EnFocus™ FSHD analysis using DLE-1 labeling and the Saphyr instrument in patients with inconclusive diagnostic Southern blot results, negative FSHD2 results, and clinically evident FSHD. Second, we performed OGM in parallel with the classical Southern blot analysis for our prospectively collected new FSHD cases. Finally, panel exome sequencing was performed to confirm the presence of FSHD2. In two patients with diagnostically inconclusive Southern blot results, OGM was able to identify shortened D4Z4 repeats on the permissive 4qA alleles, consistent with the clinical presentation. The results of the prospectively collected patients tested in parallel using Southern blotting and OGM showed full concordance, indicating that OGM is a useful alternative to the classical Southern blotting method for detecting FSHD1. In a patient showing clinical FSHD but no shortened D4Z4 repeats in the 4qA allele using OGM or Southern blotting, a likely pathogenic variant in SMCHD1 was detected using exome sequencing, confirming FSHD2. OGM and panel exome sequencing can be used consecutively to detect FSHD2.

## 1. Introduction

Facioscapulohumeral muscular dystrophy (FSHD) is an autosomal dominant disorder manifesting with atrophy and weakness of the facial, scapular, and foot dorsiflexor muscles [1]. FSHD is caused by faulty post-embryonic expression of the *DUX4* gene from the permissive 4qA haplotypes. The pathogenic process starts with the inactivation of repression mediated by D4Z4 repeats located adjacent to the *DUX4* gene [1]. Two forms of the disease are known, FSHD1 and FSHD2, which differ in their molecular pathways but are indistinguishable clinically. In the case of FSHD1, which represents ~95% of FSHD cases, the de-repression of *DUX4* is mediated through the D4Z4 contraction on the permissive 4qA haplotype. The contraction is currently considered to be fully penetrant at ≤9 repeat units of the D4Z4 locus on the permissive haplotype, whereas 10–11 repeat units of the D4Z4 locus on the permissive haplotype are considered as reduced-penetrance alleles [1]. In the remainder, ~5% of patients with FSHD2, the de-repression of *DUX4* is mediated by the hypomethylation of the D4Z4 repeats on the permissible 4qA haplotype, with some known causes of this hypomethylation being pathogenic variants in *SMCHD1, DNMT3B*, or *LRIF1* genes [1,2].

According to the current diagnostic guidelines from 2015 [2], the detection of repeated contractions causing FSHD1 is routinely performed using restriction fragment digestion of peripheral blood lymphocyte DNA and Southern blot analysis using specific probes. FSHD2 can be diagnostically confirmed in individuals who possess at least one 4qA permissive allele containing unshortened but hypomethylated D4Z4 repeats, allowing for faulty expression of the *DUX4* gene [3]. The D4Z4 hypomethylation is mediated by the insufficient expression of *SMCHD1*, *DNMT3B* or *LRIF1* gene, and pathogenic single nucleotide variants (SNV), exon duplications, or deletions in these genes can be detected using NGS, MLPA, and qPCR methods [4,5].

Southern blot analysis is a time and labor—intensive method that can be affected by SNV and CNVs adjacent to restriction enzyme sites as well as by the similarity of the D4Z4 repeats located on chromosome 10 [6,7,8,9]. Therefore, several novel methods, such as molecular combing and optical genome mapping (OGM), have been evaluated for the detection of D4Z4 contraction and have recently been reported to be an accurate and reproducible diagnostic method for FSHD1 D4Z4 repeat contraction detection [8,9,10,11,12,13]. Implementation of novel methods in clinical diagnostics requires repeatability, replicability, and reproducibility. The latter must be performed through independent validation. Therefore, our aim was to examine whether OGM can resolve our past cases with diagnostically inconclusive Southern blot results and to prospectively evaluate routine OGM performance in clinical diagnostics using a case-series comparison.

## 2. Materials and Methods

### 2.1. Patients

The study was conducted in accordance with the principles of the Declaration of Helsinki and was approved by the Institutional Review Board of the University Medical Center Ljubljana (grant 20210031). Informed consent for clinical testing, allowing the use of anonymized samples for the development and improvement of clinical genetic testing, was obtained from the individuals included in the study during their clinical examination.

A total of 23 subjects were included in the study and tested using OGM, including three patients with FSHD phenotype and inconclusive Southern blot results and negative results for FSHD2, and 20 prospectively collected asymptomatic relatives or suspected FSHD patients. The prospectively collected patients consisted of routine cases referred for FSHD testing to the Clinical Institute for Genomic Medicine (CIGM), University Medical Center Ljubljana, Slovenia, and asymptomatic family members of index patients with inconclusive Southern blot results. All samples were collected between 2021–2023. For families in which the index case had an inconclusive Southern blot, only OGM was performed on the asymptomatic family members.

### 2.2. Southern Blot

Southern blot analyses were commercially performed by Leids Universitair Medisch Centrum, Leiden, The Netherlands (LUMC), as described on their website (https://www.lumc.nl/siteassets/over-het-lumc/afdelingen/klinische-genetica/bestanden/lab/specifieke-informatie-uitvoer-diagnostiek/moleculair-genetische-diagnostiek---spierdystrofieen.pdf, accessed on 1 October 2023).

Briefly, the DNA of the patient was digested using restriction enzymes EcoRI (GAATTC)/BlnI (CCTAGG)/ApoI (RAATTY), followed by Southern blotting and hybridization using locus-specific p13E-11 probes [14,15,16]. When needed, the 4qA allele was measured using restriction fragment length digestion, Southern blotting, and locus-specific 4qA and 4qB probes. The known limitations of this method include the intactness of the p13E-11 probe DNA region at the 4q35 locus and the somatic mosaicism of the short 4q35 D4Z4 repeats. The reported results were obtained in the English language and yielded a repeat size of 4q35 D4Z4 +/−1 repeat units.

### 2.3. Sequencing Analyses

Patients with an inconclusive Southern blot result and/or suspected FSHD2 were evaluated for possible causative pathogenic variants in the *SMCHD1* gene using sequence analysis of exons 1–48, commercially performed by Leids Universitair Medisch Centrum, Leiden, The Netherlands (LUMC), as described on their website (https://www.lumc.nl/siteassets/over-het-lumc/afdelingen/klinische-genetica/bestanden/lab/specifieke-informatie-uitvoer-diagnostiek/moleculair-genetische-diagnostiek---spierdystrofieen.pdf, accessed on 1 October 2023).

Additionally, exome sequencing was performed at the CIGM using a muscular dystrophy gene panel containing 211 genes, including *SMCHD1*, *DNMT3B,* and *LRIF1*, as previously described [17].

### 2.4. Optical Genome Mapping

OGM was performed at the Clinical Institute of Genomic Medicine, University Medical Centre, Ljubljana, Slovenia. Briefly, for each sample, 1.5 million WBC from EDTA-collected whole blood were used to purify ultra-high molecular weight (HMW) DNA using the SP Blood & Cell Culture DNA Isolation Kit, following the manufacturer’s instructions (Bionano Genomics, San Diego, CA, USA). DNA was extracted from fresh blood using the ‘Bionano Prep SP Fresh Human Blood DNA Isolation Protocol’ (Document Number: 30258) and from −80 °C frozen whole blood according to the ‘Bionano Prep SP Frozen Human Blood DNA Isolation Protocol’ (Document Number: 30246) [18].

The next day, HMW DNAs was labeled using the DLE-1 enzyme (CTTAAG) and DLS (Direct Label and Stain) DNA Labeling Kit (Bionano Genomics), according to the Bionano Prep Direct Label and Stain (DLS) protocol (Document number: 30206), as previously described [18,19]. Between 4 and 12 ng/µL of HMW DNA solution per sample was loaded onto a three-flow cell Saphyr G2.3 chip and scanned on the Saphyr instrument (Bionano Genomics). Saphyr chips were run to reach a minimum yield of 500 Gbp/sample, with human GRCh38 as a reference. Molecular quality reports (MQR) were checked for adequate quality parameters and, afterward, the EnFocus FSHD pipeline was used to determine the length of the D4Z4 repeats on the chr4 and chr10 haplotypes. ICS version 5.2.21307.1 was used for processing on the Saphyr instrument. EnFocus™ FSHD Analysis was run on the 1.7.1.1. version of Access and Solve version 3.7_03302022_283 was used on either the in-house server or the compute on-demand option. The same versions of the software were used for all the analyses performed.

## 3. Results

Thirteen patients with clinically evident FSHD, one clinically suspected FSHD patient, and nine asymptomatic relatives of FSHD patients were included in the study (Table 1). The asymptomatic relatives were from several families (Table 1 and Table 2). For families in which the index cases had inconclusive Southern blot results, only OGM was performed for the subsequent asymptomatic family members. In total, Southern blotting was performed for twelve individuals, and OGM was performed for all individuals included in the study. Full data from the OGM DNA isolation, labelling, imaging, and assembly quality control steps are available in Appendix A.

In one of the three patients with initially inconclusive Southern blot results for FSHD1 and negative results for FSHD2 testing, the repeated Southern blot performed at the LUMC correctly identified the shortened size of the repeat. In the remaining two patients with clinically apparent FSHD and inconclusive Southern blot results, this was either due to the lack of a standard shortened 4q35 D4Z4 repeat, indicating the possible presence of an alternative 4qA allele (P01) or the observation of a hybrid restriction enzyme band (P02), as per the original Southern blot report.

In both patients, FSHD2 was considered and excluded after testing for the pathogenic variants of *SMCHD1*. Additional panel testing for muscular dystrophy associated genes, with exome sequencing, yielded negative results. In both cases, OGM showed a shortened D4Z4 repeat, consistent with FSHD1 (Figure 1), in concordance with the observed FSHD phenotype in these individuals (Table 1). In both cases, the presence of additional SVs and CNVs was detected in the proximal chr4 region (Appendix A), which may be the reason for the inconclusive results of the Southern blot.

Of the prospectively collected patients, eight had clinically apparent FSHD, and one had suspected FSHD. Using OGM, we showed that eight patients had shortened D4Z4 repeats on the permissive 4qA allele, consistent with FSHD1, while in one patient (P17), no shortened D4Z4 allele was detected using either OGM or Southern blot on their susceptible 4qA allele. In this patient, exome sequencing performed at the CIGM, and sequence analysis of exons 1–48 of *SMCHD1* performed commercially, concordantly detected the presence of the frameshift variant c.2141dupC (p.(Ile715fs)) in the *SMCHD1* gene. This variant was classified according to the ACMG and AMP 2015 joint consensus recommendations criteria PVS1 and PM2 [20], as a likely pathogenic variant, and the diagnosis of FSHD2 was established, consistent with the clinical presentation in this patient.

In all prospectively collected samples, the results of OGM and Southern blotting were completely concordant (Table 1), with both methods reporting a resolution of ±1 D4Z4 repeats. OGM FSHD results took an average time of one week when included in a diagnostic lab workflow and provided additional information on the length of D4Z4 repeats on the non-permissive alleles (4qB, 10qA, 10qB) (Table 1), which, in our view, is helpful in subsequent testing of their asymptomatic relatives.

In all asymptomatic family members, the results of OGM for all four alleles (4qA, 4qB, 10qA, and 10qB) were consistent with allelic segregation and the alleles of the index case (Table 2). In family F01, one asymptomatic member was identified to have a shortened D4Z4 repeat in the reduced penetrance range.

The segregation of the alleles identified by the OGM was also concordant in family F03. In this family, one suspected FSHD case was confirmed (Table 2 and Figure 2).

## 4. Discussion and Conclusions

According to the 2015 guidelines, Southern blotting is defined as the routine genetic test in FSHD diagnostics [2,21], despite recognized limitations such as sequence variation affecting probe binding and restriction sites, somatic mosaicism, and hybrid alleles [8,9,22]. Other existing methods, such as molecular combing, PCR, and those examining hypomethylation have so far not replaced this method, possibly because they are also either effort- and time-consuming, demanding in terms of interpretation, or may have limited cost-effectiveness in clinical settings with low numbers of patients per year [10,12,23,24,25,26,27]. The advantage of OGM is that it can avoid some of these issues by simultaneously detecting all typical alleles on chr4 and chr10, in addition to providing information on additional FSHD—relevant regions, such as CNVs in *SMCHD1* (which were not evaluated in this study) [11,13]. In asymptomatic cases, accurate information on the size of D4Z4 repeat on additional non—permissive 4q and 10q alleles provides, in our opinion, higher confidence to FSHD—negative results.

When assessing methods for clinical diagnostic use, it is important to consider their costs and limitations. While estimating the cost of various methods was not the aim of this study, the costs of OGM and Southern blotting are comparable to the best of the authors’ knowledge. Both approaches also have limitations. The Southern blot method has several well—known limitations, described above [6,7,8,9]. On the other hand, OGM requires a special DNA extraction step in order to provide DNA molecules of adequate size and purity, as well as an imaging apparatus enabling optical genome mapping. Furthermore, the limitations of OGM include incomplete knowledge of its performance in known rare genetic occurrences, such as mosaicism and translocations between chr4 and chr10, which need to be kept in mind when interpreting results that may not be consistent with the presenting phenotype [28].

In our study, the OGM was concordant with the Southern blot results in prospectively collected cases. Additionally, we were able to resolve two additional cases of clinically evident FSHD with diagnostically inconclusive Southern blot results. In familial asymptomatic patients, OGM provided additional information on non-permissive alleles on chr4 and chr10, which segregated in an expected pattern, thus providing more confidence to the results of asymptomatic testing.

Based on our experience that OGM is a suitable alternative method to Southern blotting for routine clinical use, we incorporated OGM in the diagnostic approach to FSHD as the primary diagnostic test for FSHD1, followed by panel exome sequencing for genes involved in muscular dystrophies, including genes involved in FSHD2, as previously described [17]. In this way, we were able to successfully include OGM and exome sequencing in our diagnostic pipeline to detect a case of FSHD2.

As demonstrated by the accurate detection of contracted repeats in the permissive 4qA allele by using OGM in patients with inconclusive Southern blot results, OGM promises to improve the diagnostic outcome and accuracy of FSHD diagnostics.

## Figures and Tables

**Figure 1 genes-14-02166-f001:**
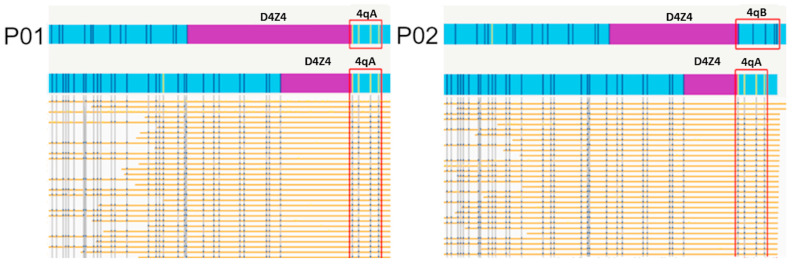
FSHD OGM maps of molecules at the 4qA allele of P01 (**left**) and P02 (**right**). Individual molecules are shown for the short 4qA haplotype assemblies, supporting the D4Z4 repeat size.

**Figure 2 genes-14-02166-f002:**
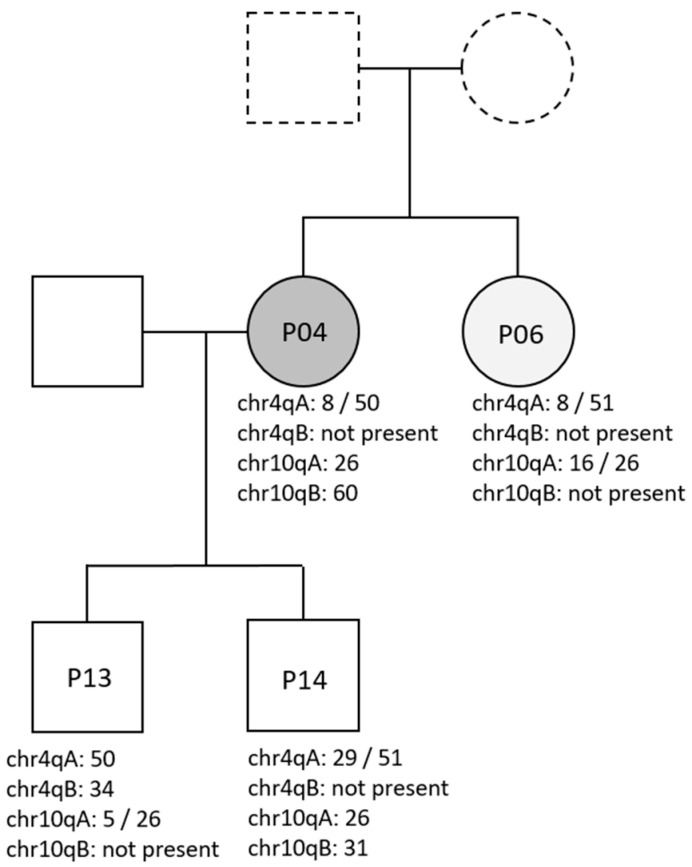
Pedigree showing segregation of chr4q and chr10q alleles in family 03. P04: Clinically apparent FSHD. P06: Suspected FSHD. P13 and P14: Asymptomatic testing. The parents of P04 and P06 were unavailable for testing.

**Table 1 genes-14-02166-t001:** Comparison between Southern blotting and optical genome mapping for FSHD testing.

Subject	Family	Indication	Southern Blot Result	OGM Result							
				4qA		4qB		10qA		10qB	
P01 *	F01	FSHD	result inconclusive	10	29			22		21	
P02 *		FSHD	result inconclusive	6		24		12	38		
P03 *	F02	FSHD	initial result inconclusive, repeated southern blot showed ~3 (+/−1) 4q35 D4Z4 repeat units	3		30		10	20		
P04	F03	FSHD	~8 (+/−1) 4q35 D4Z4 repeat units	8	50			26	60		
P05		FSHD	~8 (+/−1) 4q35 D4Z4 repeat units	7		36		2	12		
P06	F03	SF	~8 (+/−1) 4q35 D4Z4 repeat units	8	51			16	26		
P07		A	familial D4Z4 contraction not detected	56		32		15	36		
P08		A	familial D4Z4 contraction not detected			16	25	11	61		
P09		A	familial D4Z4 contraction not detected			15	50	11		27	
P10	F01	A	not performed	10		32		35		21	
P11	F01	A	not performed	29		32		34		21	
P12	F02	A	not performed	42		29		6	20		
P13	F03	A	familial D4Z4 contraction not detected	50		34		5	26		
P14	F03	A	familial D4Z4 contraction not detected	29	51			26		31	
P15		FSHD	not performed	5	50			8		17	
P16		FSHD	not performed	9		53		26	27		
P17		FSHD	No shortened 4q35 D4Z4 repeat detected	15		23/24		21	23		
P18		FSHD	not performed	7	32			22	24		
P19	F04	A	not performed	14	27			11	58		
P20		FSHD	not performed	6		15		6	22		
P21	F04	FSHD	not performed	7	27			11	45		
P22		FSHD	not performed	4		35		6	12		
P23		FSHD	not performed	7	9			19	22		

* Retrospectively included patients: FSHD—clinically diagnosed FSHD; SF—suspected FSHD; A—asymptomatic testing, relative(s) has/have FSHD; FXX—member of a particular family. Disease associated short 4qA aelles are marked in orange.

**Table 2 genes-14-02166-t002:** Allele comparison in families with FSHD.

Family	Subject	Indication	OGM Result							
			4qA		4qB		10qA		10qB	
F01	P01	FSHD	10	29			22		21	
	P10	A	10		32		35		21	
	P11	A		29	32		34		21	
F02	P03	FSHD	3		30		10	20		
	P12	A		42	29		6	20		
F03	P04	FSHD	8	50			26	60		
	P06	SF	8	51			16	26		
	P13	A		50	34		5	26		
	P14	A	29	51				26	31	
F04	P21	FSHD	7	27			11	45		
	P19	A	14	27			11	58		

FSHD—clinically diagnosed FSHD; SF—suspected FSHD; A—asymptomatic testing, relative(s) has/have FSHD; FXX—family. Disease associated short 4qA alleles are marked in orange.

## Data Availability

The authors declare that the data supporting the findings of this study are available within the paper. Original data obtained and analyzed during the current study are available from the corresponding author B.P. on reasonable request.

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
