# Peer review of "Evaluation of Optical Genome Mapping in Clinical Genetic Testing of Facioscapulohumeral Muscular Dystrophy"

_genes, 2023, doi:10.3390/genes14122166_

Round 1
Reviewer 1 Report
Comments and Suggestions for Authors
The authors have used optical genome mapping (OGM) as an alternative to the Southern blot method for detecting patients of the FSHD. The manuscript is well written and the authors have claimed that OGM is more effective than Southern blotting. The major demerit of the study is the use of known samples of FSHD patients. The authors didnot use any unknown samples of FSHD patients and compare the effectiveness between OGM and Southern blotting. My comments are provided below
1. The introduction is well written but the method sections need some improvement. For the Southern blotting procedure, the authors should provide more methodical details such as amount of DNA, sequence of probe, incubation temperature and detection method.
2. For the OGM also, the authors should describe the method in details so that it can be repeated by others.
3. For many samples in table 1, Southern blot was not performed. The authors should discuss the reason behind it.
4. The text in Figure 1 should be enlarged to make it readable.
5. The authors didnot discuss the limitation of the OGM in details. What is the difference in the cost of OGM as compared to Southern blot? What is the technical difficulties of OGM and sample processing/ purity of DNA in OGM?
Comments on the Quality of English Language
There are minor spelling mistakes in the text.
Author Response
- The introduction is well written but the method sections need some improvement. For the Southern blotting procedure, the authors should provide more methodical details such as amount of DNA, sequence of probe, incubation temperature and detection method.
We would like to answer that indeed, the study is not meant to be a direct technical comparison of laboratory performance between Southern blot vs optical genome mapping, but rather an evaluation of optical genome mapping as a diagnostic tool for routine use. For every new technology, in addition to determining the sensitivity and specificity, additional independent validation, i.e. repeatability is crucial in order for the method to be incorporated into wider use, therefore the merit of the study in our opinion is to provide an independent validation (repeatability) of the original study, in a different population and on a different set of patients.
Of course, we agree with the reviewer that it is important to provide sufficient details for the replication of the study. As we have stated in the methods, Southern blot was performed commercially at a facility that is NEN-EN-ISO 15189 accredited by the Dutch Accreditaion Council (accreditation number M007). Therefore, we have added the original references describing the Southern blot protocol in the appropriate part of the text:
'Briefly, the DNA of the patient is digested using restriction enzymes EcoRI (GAATTC)/ BlnI (CCTAGG)/ ApoI (RAATTY), followed by Southern blotting and hybridization using locus-specific p13E-11 probes [1–3].'
- Wright, T.J.; Wijmenga, C.; Clark, L.N.; Frants, R.R.; Williamson, R.; Hewitt, J.E. Fine Mapping of the FSHD Gene Region Orientates the Rearranged Fragment Detected by the Probe p13E-11. Hum Mol Genet 1993, 2, 1673–1678, doi:10.1093/hmg/2.10.1673.
- Jardine, P.E.; Koch, M.C.; Lunt, P.W.; Maynard, J.; Bathke, K.D.; Harper, P.S.; Upadhyaya, M. De Novo Facioscapulohumeral Muscular Dystrophy Defined by DNA Probe p13E-11 (D4F104S1). Archives of Disease in Childhood 1994, 71, 221–227, doi:10.1136/adc.71.3.221.
- Bakker, E.; Wijmenga, C.; Vossen, R.H.; Padberg, G.W.; Hewitt, J.; van der Wielen, M.; Rasmussen, K.; Frants, R.R. The FSHD-Linked Locus D4F104S1 (p13E-11) on 4q35 Has a Homologue on 10qter. Muscle Nerve Suppl 1995, 2, S39-44.
2. For the OGM also, the authors should describe the method in details so that it can be repeated by others.
Thank you for pointing this out. We have expanded our description of the method to include all relevant protocols:
OGM was performed at the Clinical Institute of Genomic Medicine, University Medical Centre Ljubljana, Ljubljana, Slovenia. Briefly, for each sample, 1.5 million WBC from EDTA collected whole blood were used to purify ultra-high molecular weight (HMW) DNA using the SP Blood & Cell Culture DNA Isolation Kit following manufacturer instructions (Bionano genomics, San Diego USA). DNA was extracted from fresh blood using the ‘Bionano Prep SP Fresh Human Blood DNA Isolation protocol’ (Document Number: 30258), and from -80°C frozen whole blood according to the ‘Bionano Prep SP Frozen Human Blood DNA Isolation Protocol (Document Number: 30246)[4].
The next day, HMW DNAs were labeled using the DLE-1 enzyme (CTTAAG) and DLS (Direct Label and Stain) DNA Labeling Kit (Bionano genomics), according to the Bionano Prep Direct Label and Stain (DLS) Protocol (Document number: 30206), as previously described [4,5]. Between 4 and 12 ng/ul of HMW DNA solution per sample was loaded on a three-flow cell Saphyr G2.3 chip and scanned on the Saphyr instrument (Bionano genomics). Saphyr chips were run to reach a minimum yield of 500 Gbp / sample with human GRCh38 as a reference. Molecule quality reports (MQR) were checked for adequate quality parameters and after this EnFocus FSHD specific pipeline was used to determine the length of the D4Z4 repeats and assign their haplotype on chr4 and chr10. The Saphyr instrument ran the ICS version 3. ICS 5.2.21307.1 for image processing. EnFocus™ FSHD Analysis was run on 1.7.1.1. version of Access and Solve3.7_03302022_283 versions was used on either the in-house server or on the compute on-demand option. The same versions of the software were used for all of the analyses performed.
- Bionano Support Documentation. https://bionano.com/support-documentation/
- Rogac, M.; Kovanda, A.; Lovrečić, L.; Peterlin, B. Optical Genome Mapping in an Atypical Pelizaeus-Merzbacher Prenatal Challenge. Front. Genet. 2023, 14, 1173426, doi:10.3389/fgene.2023.1173426.
- For many samples in table 1, Southern blot was not performed. The authors should discuss the reason behind it.
As we have stated in the results, for families where the index cases had inconclusive Southern blot results, only OGM was performed in the subsequent asymptomatic family members. After the publication of Vance et al., validating the use of OGM for testing of FSHD, we proceed with OGM testing only and have since testedby Southern blot, out of abundance of caution, only P17 who had similar allele lengths on all alleles of chr4 and chr10. The results of testing were concordant in that case also.
- The text in Figure 1 should be enlarged to make it readable.
As the Reviewer has suggested, we have corrected the figure to make the text readable.
- The authors didnot discuss the limitation of the OGM in details. What is the difference in the cost of OGM as compared to Southern blot? What is the technical difficulties of OGM and sample processing/ purity of DNA in OGM?
We have mentioned that OGM needs to be interpreted with caution because it may, like all other methods be affected by mosaicism, t(4;10), etc. but as the Reviewer has suggested, we have reorganized and expanded the limitations in the discussion section. Regarding the difference in the cost, we would like to comment that this was not the aim of our study. In our specific setting, OGM is more cost-effective, but this may not be the case for laboratories that have a Southern blot setup inhouse. The expanded limitation section is the following:
When assessing methods for clinical diagnostic use, it is important to consider cost and limitations. While estimating the cost of various methods was not the aim of the study, the costs of OGM and Southern blot are, to the best of the authors' knowledge, comparable. Both approaches also have their limitations - the Southern blot method has several well-known limitations, is time and labor-intensive and requires a large amount of DNA[1–4]. On the other hand, OGM requires a special DNA extraction step in order to provide DNA molecules of adequate size and purity as well as imaging apparatus enabling optical genome mapping. Furthermore the limitations of OGM include incomplete knowledge of its performance in known rare genetic occurrences such as mosaicism and translocations between chr4 and chr10, that need to be kept in mind when interpreting results that may not be consistent with the presenting phenotype [5].
- Lemmers, R.J.L.F.; Osborn, M.; Haaf, T.; Rogers, M.; Frants, R.R.; Padberg, G.W.; Cooper, D.N.; van der Maarel, S.M.; Upadhyaya, M. D4F104S1 Deletion in Facioscapulohumeral Muscular Dystrophy: Phenotype, Size, and Detection. Neurology 2003, 61, 178–183, doi:10.1212/01.wnl.0000078889.51444.81.
- Tawil, R.; van der Maarel, S.; Padberg, G.W.; van Engelen, B.G.M. 171st ENMC International Workshop: Standards of Care and Management of Facioscapulohumeral Muscular Dystrophy. Neuromuscul Disord 2010, 20, 471–475, doi:10.1016/j.nmd.2010.04.007.
- Rieken, A.; Bossler, A.D.; Mathews, K.D.; Moore, S.A. CLIA Laboratory Testing for Facioscapulohumeral Dystrophy: A Retrospective Analysis. Neurology 2021, 96, e1054–e1062, doi:10.1212/WNL.0000000000011412.
- Lemmers, R.J.L.F.; O’Shea, S.; Padberg, G.W.; Lunt, P.W.; van der Maarel, S.M. Best Practice Guidelines on Genetic Diagnostics of Facioscapulohumeral Muscular Dystrophy: Workshop 9th June 2010, LUMC, Leiden, The Netherlands. Neuromuscul Disord 2012, 22, 463–470, doi:10.1016/j.nmd.2011.09.004.
- Lemmers, R.J.L.F.; van der Vliet, P.J.; Blatnik, A.; Balog, J.; Zidar, J.; Henderson, D.; Goselink, R.; Tapscott, S.J.; Voermans, N.C.; Tawil, R.; et al. Chromosome 10q-Linked FSHD Identifies DUX4 as Principal Disease Gene. J Med Genet 2022, 59, 180–188, doi:10.1136/jmedgenet-2020-107041.
Comments on the Quality of English Language
There are minor spelling mistakes in the text.
We thank the reviewer for helping us improve our manuscript. We have re-checked the manuscript for grammar.
Reviewer 2 Report
Comments and Suggestions for Authors
The manuscript by Kovanda et al describes the use of optical genome mapping of FSHD D4Z4 repeats using EnFocus analysis and Saphyre to resolve past cases that had inconclusive Southern data. It is important for treatment and future therapeutic development to be able to accurately diagnose FSHD1/2 and this new technique of OGM is proving superior to the previous Southern blotting approach. This is of high interest to the FSHD community but this study is confirmatory to the much larger Stence et al, J. Mol Diagnostics (2021) study and a current medXriv manuscript that essentially did the same analyses.
Tables 1 and 2 the data as presented is not presented in a clear fashion and would benefit from a few lines to delineate the different allele data.
Author Response
Comments and Suggestions for Authors
The manuscript by Kovanda et al describes the use of optical genome mapping of FSHD D4Z4 repeats using EnFocus analysis and Saphyre to resolve past cases that had inconclusive Southern data. It is important for treatment and future therapeutic development to be able to accurately diagnose FSHD1/2 and this new technique of OGM is proving superior to the previous Southern blotting approach. This is of high interest to the FSHD community but this study is confirmatory to the much larger Stence et al, J. Mol Diagnostics (2021) study and a current medXriv manuscript that essentially did the same analyses.
We would like to answer that indeed, our study is not meant to be a direct technical comparison of laboratory performance between Southern blot vs optical genome mapping, but rather an evaluation of optical genome mapping as a diagnostic tool for routine clinical use. For every new technology, in addition to determining the sensitivity and specificity, additional independent validation, i.e. repeatability is crucial in order for the method to be incorporated into wider use, therefore the merit of the study in our opinion is to provide an independent validation (repeatability) of the original study, in a different population and on a different set of patients.
Tables 1 and 2 the data as presented is not presented in a clear fashion and would benefit from a few lines to delineate the different allele data.
Based on the reviewers comment, we have modified the tables, putting lines between the 4qA, 4qB, 10qA and 10qB alleles, hopefully making them more clear.
Round 2
Reviewer 1 Report
Comments and Suggestions for Authors
The authors have addressed all my concerns in the revised version of the manuscript. I support the publication of the manuscript.
Comments on the Quality of English LanguageMinor error in the text was detected.